# Vascular Calcification: A Passive Process That Requires Active Inhibition

**DOI:** 10.3390/biology13020111

**Published:** 2024-02-09

**Authors:** Ricardo Villa-Bellosta

**Affiliations:** 1Center for Research in Molecular Medicine and Chronic Diseases (CiMUS), Campus Vida, University of Santiago de Compostela, 15782 Santiago de Compostela, Spain; ricardo.villa@usc.es; 2Department of Biochemistry and Molecular Biology, University of Santiago de Compostela, 15782 Santiago de Compostela, Spain; 3The Health Research Institute of Santiago de Compostela (IDIS), Travesia da Choupana S/N, 15706 Santiago de Compostela, Spain

**Keywords:** vascular calcification, chronic kidney disease, phosphate, pyrophosphate

## Abstract

**Simple Summary:**

Vascular calcification is associated with cardiovascular complications due to accelerated arterial stiffening and atherosclerosis. A large body of evidence suggests that two principal processes are involved in vascular calcification: a passive process and an active process. In this review, we summarize the most important processes and risk factors involved in the complex process of vascular calcification.

**Abstract:**

The primary cause of worldwide mortality and morbidity stems from complications in the cardiovascular system resulting from accelerated atherosclerosis and arterial stiffening. Frequently, both pathologies are associated with the pathological calcification of cardiovascular structures, present in areas such as cardiac valves or blood vessels (vascular calcification). The accumulation of hydroxyapatite, the predominant form of calcium phosphate crystals, is a distinctive feature of vascular calcification. This phenomenon is commonly observed as a result of aging and is also linked to various diseases such as diabetes, chronic kidney disease, and several genetic disorders. A substantial body of evidence indicates that vascular calcification involves two primary processes: a passive process and an active process. The physicochemical process of hydroxyapatite formation and deposition (a passive process) is influenced significantly by hyperphosphatemia. However, the active synthesis of calcification inhibitors, including proteins and low-molecular-weight inhibitors such as pyrophosphate, is crucial. Excessive calcification occurs when there is a loss of function in enzymes and transporters responsible for extracellular pyrophosphate metabolism. Current in vivo treatments to prevent calcification involve addressing hyperphosphatemia with phosphate binders and implementing strategies to enhance the availability of pyrophosphate.

## 1. Introduction

Calcification is defined as the deposition of calcium (in the form of calcium phosphate crystals, including hydroxyapatite and carbonate apatite) in tissues. In hard tissues, such as bone and teeth, calcification is physiological. However, the deposition of calcium phosphate crystals in soft tissues, such as the liver, brain, and cardiovascular system, is considered ectopic or pathological calcification. In the cardiovascular system, calcification can occur in cardiac valves (heart valve calcification) and blood vessels (vascular calcification). Calcification in the aortic wall may manifest in two distinct layers: the medial layer, referred to as Monckeberg’s medial sclerosis, and the intimal layer, known as intimal calcification or the calcification of atherosclerotic plaques. In advanced atherosclerotic plaques, calcification is linked to inflammation. In contrast, medial calcification occurs in the medial layer independently of atherosclerosis or inflammation [1]. Furthermore, medial calcification takes place in the elastic region of the arteries and is linked to vascular smooth muscle cells (VSMCs). In contrast, intimal calcification is associated with both vascular smooth muscle cells and macrophages, particularly in lipid-rich areas of atheromatous plaques. Additionally, lipoproteins play a crucial pathogenetic role in both atherosclerosis and aortic valve sclerosis, primarily due to their influence on calcification.

Typically, the aortic artery, the coronary artery, and both the internal and common carotid arteries are the main blood vessels in which vascular calcification occurs [1]. Notably, detachment of a calcium phosphate crystal from these arteries can induce an episode of ischemia, mainly in the heart or brain (Figure 1). Moreover, vascular calcification is frequently associated with aging, progeria (accelerated aging), and diseases that accelerate aging, such as diabetes and end-stage chronic kidney disease (hemodialysis). Moreover, certain genetic diseases are linked to ectopic calcification, including vascular calcification. Examples of such conditions include pseudoxanthoma elasticum, generalized arterial calcification of infancy, familial idiopathic basal ganglia calcification (type 1), and diffuse idiopathic skeletal hyperostosis.

According to the World Health Organization, ischemic heart disease (ranked first) and stroke (ranked second) are the leading causes of death globally. Moreover, diabetes mellitus and kidney disease are ranked ninth and tenth, respectively. Therefore, it is not unreasonable to claim that vascular calcification is one of the most important factors for patient mortality around the world.

Different mechanisms for the pathogenesis of vascular calcification have been proposed (Figure 1). The two main mechanisms include (1) the loss of inhibitors of calcification, and (2) the dysregulation of phosphorus (and calcium) homeostasis. Other secondary mechanisms include (3) the differentiation of vascular smooth muscle cells to an osteochondrogenic phenotype, (4) the degradation of matrices, (5) circulating nucleation complexes, and (6) apoptosis [2,3]. However, the pathogenesis of vascular calcification is still far from being fully understood.

Elevated serum inorganic phosphate, known as hyperphosphatemia, constitutes a significant risk factor for vascular calcification in both hemodialysis patients and the general population [4]. Regarding this matter, various factors contribute to the proper control of phosphate homeostasis (see Figure 2), encompassing phosphate excretion by the kidneys and phosphate absorption by the intestine. The reduced excretion or enhanced absorption of phosphate can lead to a slight increase in serum phosphate levels, and this elevation is correlated with the presence of calcified vessels [5,6]. The dysregulation of phosphate homeostasis is associated with several diseases, including chronic renal disease, diabetes mellitus, hyperparathyroidism, vitamin D-related disorders (both hyper- and hypovitaminosis), and osteoporosis (Figure 3). Treatment with phosphate binders in hemodialysis patients is associated with a reduced progression of cardiovascular calcification [7].

Vascular smooth muscle cells incubated with a high phosphate concentration show the formation of calcium phosphate crystals [4]. This observation has been considered to be a consequence of an increased level of phosphate transport [8]. However, several studies have shown that phosphate transporters have a high affinity for inorganic phosphate (*K_m_* < 0.2 mmol/L) and are, therefore, saturated upon serum phosphate concentration. In addition, several studies have shown that the formation of calcium phosphate crystals is a physicochemical process that does not require cellular activity, suggesting that calcium/phosphate homeostasis has an important role [5,9,10]. For example, phosphate-induced calcification was observed both in devitalized VSMCs (fixed cells without cellular activity) [5] and in devitalized aortas (frozen/thawed aortas without cellular activity) [10]. Finally, calcium phosphate crystals formed in culture plates containing elastin or collagen incubated with a high phosphate concentration (phosphate-induced calcification) [5]. Interestingly, the calcium content of devitalized VSMCs/aortas is higher than that of normal VSMCs/aortas.

Phosphate-induced vascular calcification leads to two major consequences in terms of the fate of vascular smooth muscle cells (VSMCs). The first involves a significant transition to a bone-forming phenotype, characterized by the expression of osteochondrogenic markers, such as Runx2/Cbfa1 and BMP2, along with the loss of VSMC markers, like SM α-actin and SM22α [11,12,13]. The second consequence involves apoptosis-dependent matrix mineralization, a process observed in arteries from pediatric dialysis patients [14] as well as in cultured human VSMCs [15,16]. Several studies have shown that hydroxyapatite deposits can induce the transition of VSMCs to a bone-forming phenotype [5,9,10,17], suggesting a compensatory response to ectopic calcification [6]. Notably, VSMCs incubated with hydroxyapatite exhibit both the loss of SM22α expression and the overexpression of BMP2 in a dose-dependent manner [10].

To prevent the formation of hydroxyapatite crystals, the body produces inhibitors of calcification using adenosine-5′-triphosphate (ATP), including proteins and low-molecular-weight inhibitors. Recent work provides evidence that the purinergic system plays an important role and, in particular, has links to extracellular pyrophosphate metabolism [18]. Extracellular pyrophosphate is the main endogenous inhibitor of vascular calcification [19], which is produced via the hydrolysis of extracellular ATP [20]. Moreover, ATP directly inhibits calcium phosphate deposition [21], which is a molecular mechanism similar to pyrophosphates, bisphosphonates (nonhydrolyzable analogous to pyrophosphate), and polyphosphates [22,23,24]. Enzymes and transporters involved in extracellular ATP/pyrophosphate metabolism include alkaline phosphatase, members of the ENPP and eNTPD families, ecto-5′-nucleotidase [18], equilibrative nucleoside transporters [25], phosphate transporters [26,27], and pumps/channels that release ATP extracellularly [28,29].

## 2. Phosphate Homeostasis and Vascular Calcification

Inorganic phosphate plays a critical role in many biological processes including metabolic regulation, bioenergetics, cell proliferation, cell signaling, bone mineralization, and membrane integrity [30,31,32,33,34]. Many different daily foods contain inorganic phosphate, including rice, milk, cheese, chicken, salmon, potatoes, tomatoes, and eggs [35]. In food, inorganic phosphate is bound to organic molecules (organic phosphate), and it is released in the gastrointestinal tract by hydrolysis. Moreover, phosphate additives are present in many processed food products in the form of phosphate salts (including potassium or sodium phosphate), sodium polyphosphate, and phosphoric acid [36]. The recommended dietary daily phosphorus intake in healthy >19-year-old adults is ~700 mg per day for both women and men [37]. But this recommendation varies between 100 and 1250 mg per day depending on age, lactation, or pregnancy.

Daily phosphorus intake varies according to the consumption of dairy products and phosphate additives. For example, bakery products and chicken contribute 10% and 5%, respectively, of total phosphate intakes. In adults, the average daily phosphate intake from foods is >1000 mg for both women and men. Moreover, the bioavailability of phosphate salts contained in dietary supplements and organic phosphate contained in foods are approximately 70% and 40%–70%, respectively [37].

Following absorption, phosphate is transported across cell membranes. In extracellular fluids, phosphate undergoes one of three fates (as illustrated in Figure 2): it is (1) primarily eliminated by the kidneys, (2) transported into cells, or (3) deposited in bone or soft tissues. In healthy adults, the oral intake of phosphate is predominantly balanced by its excretion in urine and feces. The normal phosphate concentration in serum or plasma ranges from 2.5 to 4.5 mg/dL (0.81 to 1.45 mmol/L).

Phosphate reabsorption along renal proximal tubules is the principal control of phosphate homeostasis. Significantly, in normal adults, the primary daily phosphate filtered by the glomerulus is reabsorbed by the renal tubules [38]. Nevertheless, intestinal phosphate absorption can impact plasma phosphate levels. Importantly, phosphorus intake appears to be on the rise due to the increasing consumption of highly processed foods, especially fast foods and convenience foods. While high dietary intake of phosphate increases renal excretion, low dietary intake of phosphate enhances renal reabsorption. Many other factors can also affect phosphate homeostasis (Figure 3). For example, fibroblast growth factor 23, parathyroid hormone, 1.25-OH_2_-vitamin D3, glucocorticoids, dopamine, acidosis, and hypokalemia increase renal excretion of phosphate [2,3]. Moreover, renal reabsorption of phosphate is increased by 1.25-OH_2_-vitamin D3, insulin-like growth factor 1, insulin, and thyroid hormone, and during alkalosis [2,3].

Considering the vital roles of phosphate in crucial cellular processes, it is unsurprising that deviations from normal serum phosphate concentrations can lead to severe clinical disorders. Phosphate deficiency typically results in anemia, rhabdomyolysis, muscle weakness, abnormal bone mineralization, and impaired leukocyte function. However, patients with severe malnutrition and people with genetic phosphate regulation disorders can induce hypophosphatemia [39]. In addition, some medications can have an adverse effect on phosphate levels, and phosphate can interact with certain medications [40]. Moreover, several diseases are correlated with phosphate homeostasis dysregulation, including diabetes mellitus, chronic kidney disease, vitamin D-related diseases (hyper- and hypovitaminosis), osteoporosis, and hyperparathyroidism [38,41]. Notably, the increased absorption or decreased excretion of phosphate can induce a relatively small hyperphosphatemia, which is correlated with vascular calcification due to the saturation of inhibition or increased calcium phosphate crystal formation.

## 3. Passive Process: Biological Mineralization

In an aqueous system containing calcium and phosphate, phosphate is found as a free phosphate ion in solution, and according to its triprotic equilibrium, aqueous inorganic phosphate exists in four forms (Figure 4): (1) trihydrogen phosphate ion (H_3_PO_4_), (2) dihydrogen phosphate ion (H_2_PO_4_^−^; pK_a1_ = 2.1), (3) hydrogen phosphate ion (HPO_4_^2−^; pK_a2_ = 6.9), and (4) phosphate ion (PO_4_^3−^; pK_a3_ = 12.4). The phosphate ion and trihydrogen phosphate dominate under strongly basic and acidic conditions, respectively. In extracellular fluid (pH = 7.4), only H_2_PO_4_^−^ and HPO_4_^2−^ ions are present in significant amounts at a ratio of 1:4. By contrast, in the cytosol (pH = 7) and lysosomes (pH = 4.8), this ratio is inverted (1.6:1 and 99:1, respectively).

Notably, various salt types are obtained by the charge neutralization of these different inorganic phosphate ions with calcium ions [42], including (1) monocalcium phosphate anhydrous (Ca(H_2_PO_4_)_2_), (2) dicalcium phosphate anhydrous (CaHPO_4_), and (3) β-tricalcium phosphate (β-Ca_3_(PO_4_)_2_) [43]. Both Ca(H_2_PO_4_)_2_ and CaHPO_4_ are hydrated to generate monocalcium phosphate monohydrate (Ca(H_2_PO_4_)2H_2_O) and dicalcium phosphate dihydrate (CaHPO_4_2H_2_O), respectively. CaHPO_4_2H_2_O, also called brushite, is often found in calcified tissues, whereas Ca(H_2_PO_4_)_2_, Ca(H_2_PO_4_)2H_2_O, CaHPO_4_, and β-Ca_3_(PO_4_)_2_ have never been found in calcifications. Interestingly, whitlockite (the Mg-substituted β-tricalcium phosphate form) has been found in some calcified tissues, such as the aorta, in hemodialysis patients [44,45] but does not form under physiological conditions.

Crystalline hydroxyapatite (Ca_10_(PO_4_)_6_(OH)), the main component of bone and calcified tissues [46], and two of its precursors (amorphous calcium phosphate (Ca_9_(PO_4_)_6_nH_2_O) and octocalcium phosphate (Ca_8_H_2_(PO_4_)_6_5H_2_O)) are the final product of the passive aggregation of calcium phosphate salts in neutral and basic solutions [42,47]. Moreover, amorphous calcium phosphate, which consists mainly of roughly spherical Ca_9_(PO_4_)_6_ clusters (called Posner’s clusters), is also found in soft tissue pathological calcifications. This type of cluster appears to be energetically favored compared with (Ca_3_(PO_4_)) and Ca_6_(PO_4_)_4_ clusters [48]. Therefore, the structure of hydroxyapatite can be interpreted as an aggregation of Posner’s clusters [47,49]. Notably, Mg^2+^ and ATP are critical for the stabilization of amorphous calcium phosphate [50,51].

According to the charge neutralization theory of calcification proposed by Urry [52], the elevated glycine content in elastin and collagen proteins promotes the formation of β-turns, facilitating interactions with calcium ions. Therefore, calcium phosphate salts are deposited on these extracellular matrix proteins both in vitro and in vivo [5,10] in a physicochemical manner. Notably, a study showed that calcium phosphate crystals formed and grew on collagen and elastin fibers incubated with a high phosphate concentration in the absence of cells [5]. Another study showed that arterial calcification was significantly reduced in a mouse model of elastin haploinsufficiency [53].

## 4. Active Process: Synthesis of Inhibitors

Extracellular fluids, including serum, exhibit a state of supersaturation with phosphate and calcium. This leads to a natural tendency for calcium phosphate to spontaneously precipitate. Various endogenous inhibitors, both of low and high molecular weights, are present in extracellular fluids to prevent this precipitation. Low-molecular-weight inhibitors include pyrophosphate (and citrate). Moreover, high-molecular-weight inhibitors include fetuin-A, matrix Gla protein (MGP), and osteopontin [2].

The circulating plasma glycoprotein Fetuin-A was originally discovered as an inhibitor of vascular calcification [54,55]. It can bind to calcium, has anti-inflammatory properties, and plays an important role in improving hepatic free fatty acid-induced insulin resistance. Notably, knockout mice in Fetuin-A develop excessive calcification in different organs, including kidneys, testes, skin, heart, and blood vessels [55]. Studies suggest that in both kidney transplant recipients with cardiac valvular calcification [56] and hemodialysis patients with abdominal aortic calcification [57], serum fetuin-A levels are increased. By contrast, lower fetuin-A levels are associated with both basal ganglia calcification (Fahr syndrome) [58] and higher vascular calcification scores in patients with end-stage disease [59]. However, no significant correlation was found between fetuin-A levels and both valvular calcification in kidney transplant recipients [60] or calcification in hemodialysis patients [61].

MGP, an inhibitor of vascular calcification [62,63,64], contains several modified amino acid residues including gamma-carboxyglutamic acid. It is synthetized by VSMCs and chondrocytes and is a mineral-binding extracellular matrix protein [64]. Mutations of the gene encoding human MGP cause Keutel syndrome [65], and MGP-null mice exhibit spontaneous artery and cartilage calcification [64]. Studies have also reported associations between vascular calcification and plasmatic MGP in hypertensive, diabetic, atherosclerotic, and uremic patients [66]. Moreover, vitamin K is required to completely synthesize MGP [67].

The sialic-acid-rich glycoprotein known as osteopontin is purified from bones [68]. It is a non-collagenous bone matrix protein that regulates calcification. However, osteopontin phosphorylation is required to inhibit vascular smooth muscle calcification [63]. In human aortic atherosclerotic lesions, osteopontin is expressed by smooth-muscle-derived foam cells [69,70]. Also, in human coronary atherosclerotic plaques, both endothelial cells—macrophages and smooth muscle cells—synthesize osteopontin mRNA and protein [71]. Osteopontin and MGP are present in atherosclerotic lesions and upregulated in proliferating VSMCs [72,73]. The upregulation of osteopontin mRNA levels is not correlated with calcification in human vascular cells. However, high levels of MGP are correlated with calcification in vitro [74]. Finally, studies with calcified aortas suggest that osteopontin is a poor inhibitor of calcification [75,76].

Extracellular pyrophosphate is the main, and potent, inhibitor of vascular calcification, which directly inhibits the formation and growth of calcium phosphate crystals both in vitro and in vivo [19,22]. The production of endogenous pyrophosphate can prevent vascular calcification [6,77] in the normal range of plasmatic phosphate. However, endogenous pyrophosphate concentration is not sufficient to prevent calcification with hyperphosphatemia [6]. A reduction in plasmatic pyrophosphate concentration is associated with calcification [19] in patients under standard hemodialysis [78]. Also, in mouse models of pseudoxanthoma elasticum [79] and progeria [80], plasma pyrophosphate levels are decreased. Notably, both mouse models are characterized by excessive calcification, and daily injections of exogenous pyrophosphate prevent calcification [79,80]. Moreover, the therapeutic administration of pyrophosphate also reduced vascular calcification in murine models including ApoE-null mice [81] and uremic mice and rats [82,83].

## 5. Extracellular Pyrophosphate Metabolism and Vascular Calcification

The extracellular hydrolysis of ATP is the main source of extracellular pyrophosphate (Figure 5). Notably, ectonucleotide pyrophosphatase/phosphodiesterase 1–3 subfamily (eNPP) are the main enzymes involved in the production of extracellular pyrophosphate, which releases pyrophosphate and AMP from the hydrolysis of ATP [20]. However, eNPP1 is the main enzyme involved in extracellular pyrophosphate production both in the aorta and vascular smooth muscle cells [20,80]. The loss of function induced by an eNPP1 mutation is associated with generalized arterial calcification of infancy (GACI), which is characterized by the calcification of arteries [84]. In addition, eNPP1-null mice often develop artery calcification [85], and eNPP1 replacement therapy prevented vascular calcification and mortality in a mouse model of GACI [86].

The membrane-bound ecto-5′-nucletotidase (NT5E, also known as CD73) is an ectoenzyme involved in extracellular pyrophosphate metabolism (Figure 5). This enzyme hydrolyzes extracellular AMP releasing adenosine and phosphate [18]. The loss of function in CD73 is associated with medial calcification [87].

Pyrophosphate can mainly be removed to phosphate via tissue non-specific alkaline phosphatase (TNAP) in extracellular fluids. Alkaline phosphatase overexpression induces medial calcification in aortic rings ex vivo [20] and skeletal mineralization in vivo [88]. In contrast, the addition of alkaline phosphatase to culture media induces vascular smooth muscle calcification in vitro [10,20]. Notably, TNAP activity increased in uremic rats [76] and a mouse model of Hutchinson–Gilford progeria syndrome [80], with both mouse models characterized by the presence of medial calcification. Notably, the inhibition of TNAP prevented vascular smooth muscle calcification in vitro and protected against medial arterial calcification in a chronic kidney disease–mineral bone disorder mouse model [89] and a pseudoxanthoma elasticum mouse model [90,91], but not in Enpp1 mutant mice [90].

Extracellular adenosine and phosphate must be transported into cells to generate ATP by metabolic pathways, including mitochondria. The loss of function in the adenosine transporter known as “equilibrative nucleoside transporter 1” (ENT1, also called SLC29A1) is associated with “diffuse idiopathic skeletal hyperostosis” in humans, which is characterized by ectopic calcification [25]. Moreover, inorganic phosphate is transported by two families of sodium phosphate transporters (NaPi)—type II (SLC34) and type III (SLC20) [92,93]—which show high but different affinities for phosphate ions (Figure 4). The type II family (NaPi-II) comprises three members expressed in the kidneys (NaPi-IIa and NaPi-IIc) and the small intestine (NaPi-IIb), two important control sites of phosphate homeostasis. Type III sodium phosphate cotransporters include Pit-1 and Pit-2. Both mediate the transport of phosphate ions across the cell membrane and are ubiquitously expressed. The expression of Pit-1 mRNA is highest in VSMCs, osteoblasts, and bone marrow [8,27]. The expression of Pit-2 is highest in VSMCs, the heart, the brain, and the liver [27]. Loss-of-function Pit-2 mutations are associated with familial idiopathic basal ganglia calcification in humans [26], and the global knockout of Pit-2 also causes basal ganglia calcification in mice [94].

The cycle of the extracellular pyrophosphate metabolic is closed with the transport of ATP to extracellular fluids [29]. Cellular ATP release occurs through both membrane protein transport and exocytotic mechanisms [28]. Different membrane proteins have been suggested to mediate the transport of ATP to extracellular milieu, including connexin hemichannels, pannexin, and ATP-binding cassette sub-family C member 6 (ABCC6, also known as multidrug resistance-associated protein 6 (MRP6)). Recent reports have suggested that ABCC6 mediates ATP excretion in the liver [29,95]. The loss of function in ABCC6 mutation reduces plasmatic pyrophosphate concentrations [95] and is associated with *Pseudoxanthoma elasticum*, which is characterized by ectopic calcification [96,97].

In a healthy aortic wall, the synthesis of pyrophosphate from ATP hydrolysis occurs at a rate several times faster than the hydrolysis of pyrophosphate [10,20]. Vascular smooth muscle cells, the primary cell type involved in the calcification of the aortic wall, play a crucial role in preventing medial vascular calcification through the expression and activity of both TNAP and eNPP1 enzymes [20]. The expression of these enzymes changes during different stages of the calcification process [10]. During the early phase, when calcification is not yet present, there is an increase in eNPP1 activity and a decrease in TNAP activity in vascular smooth muscle cells, both in vitro, ex vivo, and in vivo. In contrast, during the late phase when calcification is established and Runx2/Cbfa1 is expressed, hydroxyapatite increases both eNPP1 and TNAP activities [10].

Moreover, the calcification of atherosclerotic plaques also entails the accumulation of macrophages within the artery wall. Macrophages exhibit remarkable plasticity and can alter their physiology and function in response to environmental signals [98,99]. Atherosclerotic lesions contain cells expressing markers of classical macrophages (M1 macrophages) that promote inflammation, inhibit cell proliferation, and cause tissue damage. Furthermore, atherosclerotic lesions contain cells expressing markers of alternative macrophages (M2 macrophages) that promote cell proliferation and tissue repair. Importantly, M2 macrophages release more ATP and increase pyrophosphate synthesis through elevated eNPP1 expression and activity, as compared to M1 macrophages, suggesting an anti-calcifying property [100]. In addition, phosphate-induced macrophages (MPi) express M2 markers and exhibit similar anti-calcifying properties [101].

## 6. Role of Lipoproteins in Atherosclerosis and Aortic Valve Sclerosis

Finally, atherosclerosis is a condition where atherosclerotic lesions, consisting of cholesterol, fatty substances, cellular waste products, calcium, and fibrin, build up within the arteries. Low-density lipoprotein (LDL) cholesterol is a key player in atherosclerosis. Elevated levels of LDL can lead to the deposition of cholesterol in the arterial walls, triggering an inflammatory response. Lipoproteins, especially LDL, contribute to the formation of atherosclerotic plaques. The interaction between lipoproteins and inflammatory processes can also influence the calcification of these plaques over time [102]. Moreover, aortic valve sclerosis involves the thickening and calcification of the aortic valve, impeding its proper function. Like atherosclerosis, lipoproteins, particularly LDL cholesterol, are implicated in pathogenesis. Elevated LDL levels can contribute to the deposition of cholesterol in valve tissues. The presence of lipoproteins in valve tissues can initiate inflammatory responses and contribute to the calcification process, leading to the hardening and dysfunction of the aortic valve [103]. In both conditions, there is a common thread of inflammation. Inflammatory responses are triggered by the presence of lipoproteins, and inflammation is closely linked to the calcification of vascular and valvular tissues. The interplay between lipoproteins, inflammatory cells, and various signaling pathways contributes to the deposition of calcium salts in the affected tissues [104]. Understanding the role of lipoproteins in the pathogenesis of atherosclerosis and aortic valve sclerosis has significant clinical implications [105]. Strategies to manage lipoprotein levels, particularly LDL cholesterol, through lifestyle modifications and medications, are crucial for preventing and managing these cardiovascular conditions. Addressing the inflammatory component is also a potential avenue for therapeutic interventions. In conclusion, the involvement of lipoproteins in atherosclerosis and aortic valve sclerosis underscores their significance in the pathogenesis of these conditions, particularly in influencing the processes of inflammation (active process) and passive calcification. This knowledge is valuable for developing targeted approaches for prevention and treatment [102,105].

## 7. Conclusions

Several studies suggest two principal processes involved in vascular calcification: a passive process and an active process. The formation and deposition of calcium phosphate crystals, mainly in the form of hydroxyapatite, is a passive process that does not require cellular activity and in which hyperphosphatemia is a critical risk factor. However, the synthesis of calcification inhibitors requires the active cellular synthesis of ATP. In relation to this, two main mechanisms are involved in the active process of calcification (Figure 6): the impairment of phosphate homeostasis (which leads to hyperphosphatemia) and the loss of inhibitor synthesis (which mainly leads to pyrophosphate deficiency).

The severity of calcification does not correlate with the circulating pyrophosphate concentration. This observation suggests that there needs to be an appropriate balance between phosphate and pyrophosphate concentrations [1]. Therefore, there is a need for further investigation into phosphate and pyrophosphate homeostasis to aid in the development of innovative therapeutic approaches for ectopic calcification in cardiovascular structures. These approaches may involve strategies to mitigate hyperphosphatemia [7], the administration of exogenous pyrophosphate [80,81,82,83], and strategies to increase pyrophosphate concentrations by targeting the ATP/pyrophosphate metabolic cycle [106,107]. In relation to this, the inhibition of TNAP [89,90,91] and the exogenous administration of eNPP1 [86] prevent vascular calcification.

In clinical practice, it is crucial to assess phosphate and pyrophosphate homeostasis by evaluating both plasmatic phosphate and pyrophosphate levels. When elevated phosphate levels are detected in the blood, the initial therapeutic strategies to prevent vascular calcification should include the administration of phosphate binders to reduce circulating phosphate levels and address any dysregulated phosphate homeostasis if present. Furthermore, in cases of low pyrophosphate levels, therapeutic strategies should involve the administration of exogenous pyrophosphate and interventions to enhance the availability of endogenous pyrophosphate.

## Figures and Tables

**Figure 1 biology-13-00111-f001:**
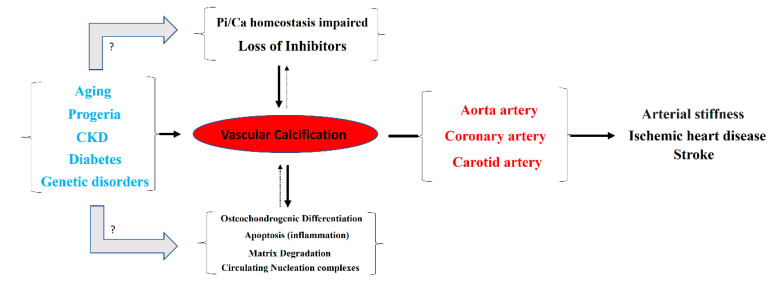
Overview of the pathogenesis and consequences of vascular calcification.

**Figure 2 biology-13-00111-f002:**
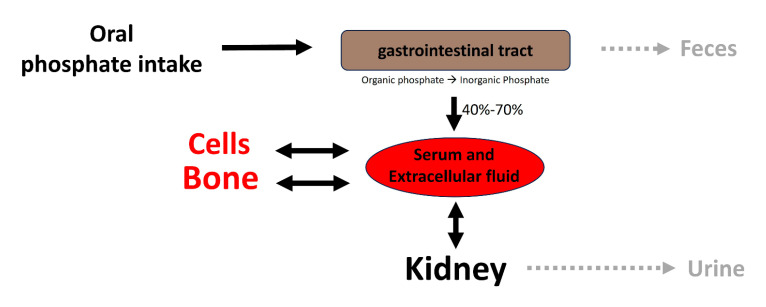
Overview of phosphate homeostasis. Extracellular and plasma phosphate balance is a complex process of flux between body compartments.

**Figure 3 biology-13-00111-f003:**
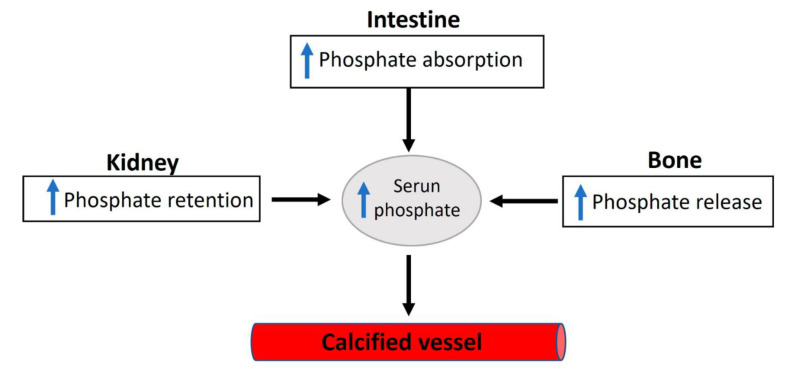
An overview of the factors involved in dysregulated phosphate homeostasis.

**Figure 4 biology-13-00111-f004:**
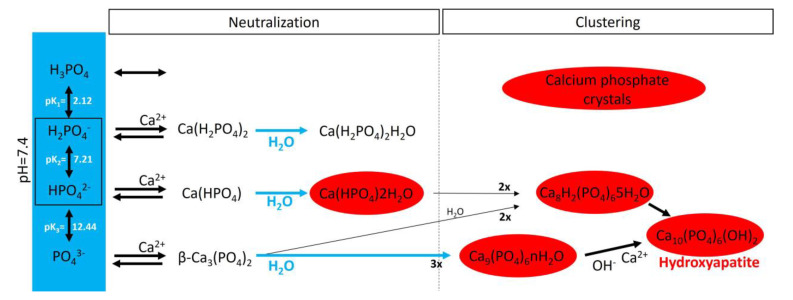
A schematic representation of calcium phosphate crystal formation. The blue box shows the four species of inorganic phosphate. In the red circles are the calcium phosphate crystals found in bone and calcified tissues, including hydroxyapatite (Ca_10_(PO_4_)_6_(OH)_2_), amorphous calcium phosphate (Ca_9_(PO_4_)_6_nH_2_O), octocalcium phosphate (Ca_8_H_2_(PO_4_)_6_5H_2_O), and dicalcium phosphate dihydrate (Ca(HPO_4_)2H_2_O).

**Figure 5 biology-13-00111-f005:**
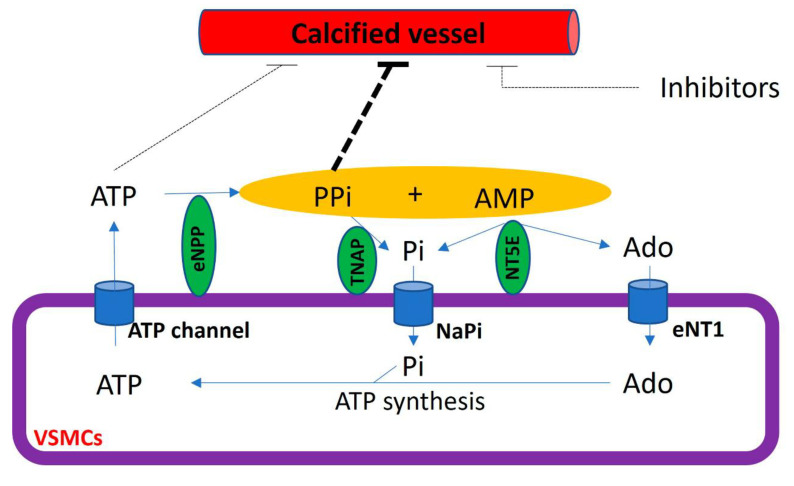
Schematic representation of the extracellular pyrophosphate metabolism. Adenosine-5′ triphosphate (ATP). Adenosine-5′ monophosphate (AMP). Pyrophosphate (PPi). Adenosine (Ado). Inorganic phosphate (Pi). Tissue non-specific alkaline phosphatase (TNAP). Ectonucleotide pyrophosphatase phosphodiesterase (eNPP). Ecto-5′ nucletotidase (NT5E). Equilibrative nucleoside transporter 1 (eNT1). Sodium phosphate transporter (NaPi).

**Figure 6 biology-13-00111-f006:**
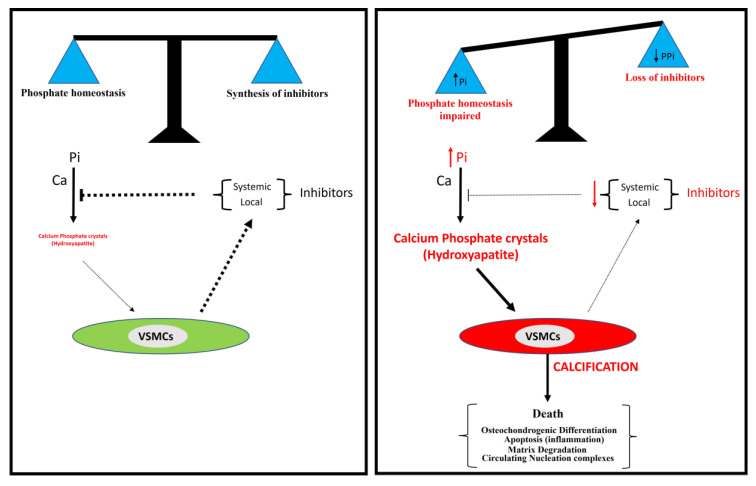
A model for vascular calcification. Vascular calcification depends mainly on the balance between the concentration of inorganic phosphate (Pi) and the synthesis of inhibitors, including pyrophosphate. The loss of synthesis of calcification inhibitors or elevated phosphate concentration in serum (hyperphosphatemia) is associated with the excessive accumulation of calcium phosphate crystals in the aortic wall, mainly in the form of hydroxyapatite.

## Data Availability

Not applicable.

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
