# Peer review of "Vascular Calcification: A Passive Process That Requires Active Inhibition"

_biology, 2024, doi:10.3390/biology13020111_

Round 1

Reviewer 1 Report

Comments and Suggestions for Authors

In this review, Villa-Belosta summarizes the processes and risk factors involved in vascular calcification. The article is well written providing a helpful differentiation between active and passive processes involved in calcification. However, two major points need to be addressed:

1.     Both atherosclerosis and aortic valve sclerosis involve lipoproteins playing a crucial pathogenetic role, not last because of their impact on calcification. This aspect and its integration into the concept of active and passive processes should be considered.

2.     Upon reviewing the similarity report of this paper, a total similarity of 45% was noted. While it can be challenging to avoid such overlap in particular in a review article, efforts should be made to significantly reduce it through appropriate rephrasing.

Minor points

1.     Line 96 and 97: spelling of „25-hydroxyvitamin“ and „hydroxylase“

2.     Line 199: Do not provide title („Professor“), that´s unusual

3.     Line 240 to 242: „Although … osteopontin … high levels of MGP“: I do not understand the link in this sentence.

4.     Line 246 to 248: spelling „In“ and „however“

Comments on the Quality of English Language

Minor editing (spelling) of English language required

Author Response

In this review, Villa-Belosta summarizes the processes and risk factors involved in vascular calcification. The article is well written providing a helpful differentiation between active and passive processes involved in calcification. However, two major points need to be addressed:

  1. Both atherosclerosis and aortic valve sclerosis involve lipoproteins playing a crucial pathogenetic role, not last because of their impact on calcification. This aspect and its integration into the concept of active and passive processes should be considered.

Response: This author would like to thank the reviewer for his comments. Additional information has been included in the reviser form according to the reviewer suggestion (new section 6). 

  1. Upon reviewing the similarity report of this paper, a total similarity of 45% was noted. While it can be challenging to avoid such overlap in particular in a review article, efforts should be made to significantly reduce it through appropriate rephrasing.

Response: Manuscript has been improved to reduced similarity. More than 41 references shown 1% (<45 words) of similarity according with the similarity report. In addition, many definitions are identified as similar when they really are not. For example: “ATP-binding cassette sub-family C member 6”; “Hutchinson–Gilford progeria syndrome”; “equilibrative nucleoside transporter 1”;  “diffuse idiopathic skeletal hyperostosis”; or “Conflicts of Interest: The authors declare that the research was conducted in the absence of any commercial or financial relationships that could be construed as a potential conflict of interest”.

Minor points

  1. Line 96 and 97: spelling of „25-hydroxyvitamin“ and „hydroxylase“
  2. Line 199: Do not provide title („Professor“), that´s unusual
  3. Line 240 to 242: „Although … osteopontin … high levels of MGP“: I do not understand the link in this sentence.
  4. Line 246 to 248: spelling „In“ and „however“

Response: This author would also like to acknowledge the identification of these minor errors. All of them have been improved.

Reviewer 2 Report

Comments and Suggestions for Authors

The present review depicts molecular pathways implicated in physiological anticalcification process. It is clear and gives an apercu of phosphate homeostasis and deficiencies of calcification inhibitors with a specific focus on pyrophosphate. I would have the following remarks.

There is quite important number of self citations (17/92) according to the journal policies these references could be changed for more appropriated ones. As for example:

The resolution of the figures is poor and sometimes these are hard to read

P1L43: aorta instead of aortic artery. In figure 1 the sens of Aorta(iliac) is not clear.

P2L45: when writing “detachment of a calcium phosphate crystal from these arteries can induce an episode of ischemia the author may reefer to thrombotic embolization in the downstream circulation. To my knowledge this is caused by platelet and platelet/leucocytes aggregates (activated by the prothrombotic material that is released by plaque rupture) rather than by the obliteration of vessels by hydroxyapatite itself.

L125: ” legumes, vegetables” the nuance is not obvious

L247: capital letter to “However”

L273: “The membrane-bound ecto-5’-nucletotidase (also known as CD73) is another ectoenzyme involved in extracellular pyrophosphate metabolism” this formulation is not correct as CD73 does not metabolize PPi.

L276: CD39 instead of CD73

L277: “Pyrophosphate is mainly degraded to phosphate…” mainly can be removed

L285-286 “a pseudoxanthoma elasticum mouse model[85], and a Abcc6-/- mouse model[86]” these models are the same.

L305: “transport of to the extracellular fluids[3].”

Comments on the Quality of English Language

no comments

Author Response

The present review depicts molecular pathways implicated in physiological anticalcification process. It is clear and gives an apercu of phosphate homeostasis and deficiencies of calcification inhibitors with a specific focus on pyrophosphate. I would have the following remarks.

There is quite important number of self citations (17/92) according to the journal policies these references could be changed for more appropriated ones. As for example:

Response: This author would like to thank the reviewer for his comments. Some self-citation has been deleted in the reviser form of this manuscript. In the revised form there are 14 self-citations of 107 references (13%). However, all these self-citations are important. 

The resoltion of the figures is poor and sometimes these are hard to read

 Response: The resolution of the figures has been improved.

P1L43: aorta instead of aortic artery. In figure 1 the sens of Aorta(iliac) is not clear.

P2L45: when writing “detachment of a calcium phosphate crystal from these arteries can induce an episode of ischemia” the author may reefer to thrombotic embolization in the downstream circulation. To my knowledge this is caused by platelet and platelet/leucocytes aggregates (activated by the prothrombotic material that is released by plaque rupture) rather than by the obliteration of vessels by hydroxyapatite itself.

L125: ” legumes, vegetables” the nuance is not obvious

L247: capital letter to “However”

L273: “The membrane-bound ecto-5’-nucletotidase (also known as CD73) is another ectoenzyme involved in extracellular pyrophosphate metabolism” this formulation is not correct as CD73 does not metabolize PPi.

L276: CD39 instead of CD73

L277: “Pyrophosphate is mainly degraded to phosphate…” mainly can be removed

L285-286 “a pseudoxanthoma elasticum mouse model[85], and a Abcc6-/- mouse model[86]” these models are the same.

L305: “transport of to the extracellular fluids[3].”

Response: This author would also like to acknowledge the identification of these minor errors. All of them have been improved.

Round 2

Reviewer 1 Report

Comments and Suggestions for Authors

The authors has adequately addressed the issues raised by this reviewer. 

One more minor point: The term "plaque" (line 355) should be replaced by "atherosclerotic lesions".

Comments on the Quality of English Language

 Minor editing of English language is still required.

Author Response

The term plaque has been replaced by “atherosclerotic lesions”. Thank you very much for this comment.